# FlashPCR: Revolutionising qPCR by Accelerating Amplification through Low ∆T Protocols

**DOI:** 10.3390/ijms25052773

**Published:** 2024-02-28

**Authors:** Stephen A. Bustin, Sara Kirvell, Tania Nolan, Gregory L. Shipley

**Affiliations:** 1Medical Technology Research Centre, Faculty of Health, Medicine and Social Care Anglia, Ruskin University, Chelmsford CB1 1PT, UK; sara.kirvell1@aru.ac.uk (S.K.); tanianolan@btinternet.com (T.N.); 2Shipley Consulting, Vancouver, WA 98682, USA; gshipley14@me.com

**Keywords:** reverse transcription, qPCR, COVID-19, molecular diagnostics, point of care

## Abstract

Versatility, sensitivity, and accuracy have made the real-time polymerase chain reaction (qPCR) a crucial tool for research, as well as diagnostic applications. However, for point-of-care (PoC) use, traditional qPCR faces two main challenges: long run times mean results are not available for half an hour or more, and the requisite high-temperature denaturation requires more robust and power-demanding instrumentation. This study addresses both issues and revises primer and probe designs, modified buffers, and low ∆T protocols which, together, speed up qPCR on conventional qPCR instruments and will allow for the development of robust, point-of-care devices. Our approach, called “FlashPCR”, uses a protocol involving a 15-second denaturation at 79 °C, followed by repeated cycling for 1 s at 79 °C and 71 °C, together with high Tm primers and specific but simple buffers. It also allows for efficient reverse transcription as part of a one-step RT-qPCR protocol, making it universally applicable for both rapid research and diagnostic applications.

## 1. Introduction

In addition to its central status as a research technique, the real-time polymerase chain reaction (qPCR) [1,2] has found extensive applied uses in clinical [3], veterinary [4] and agricultural [5] diagnostics, as well as public health surveillance [6,7]. However, as molecular diagnostic applications continue to expand and evolve, traditional qPCR protocols and assay designs present inherent challenges for rapid results and point-of-care (PoC) applications. One particular limitation is the time it takes for qPCR instruments to ramp up and down the 35 °C temperature differential necessitated by the standard 95 °C denaturation/60 °C polymerisation protocol. There has been a focus on eliminating heating blocks, for example, through the use of microfluidic qPCR chips [8,9], continuous flow PCR [10,11] and plasmonic nanoparticles [12,13], but there has been a surprising lack of adjustment to the original protocols. This is despite the introduction of improved enzymes, reagents, and plasticware. Even extreme PCR protocols [14,15,16] use high denaturation and low annealing temperatures. In practice, this means that, in combination with long dwelling times, a typical 40-cycle qPCR assay still takes 45 min to complete. This approach is clearly unsuitable for PoC applications, particularly in infectious disease diagnosis, where quick and accurate identification of the causative agent is crucial for effective treatment and containment. Additionally, in fields such as environmental monitoring [17], food safety [18], and forensic analysis [19], reliable and rapid detection is crucial for timely decision-making and intervention. Whilst sample preparation remains a bottleneck within the PCR workflow, this issue is being addressed [20,21,22,23,24,25,26], making it even more important to address the PCR step itself.

This study addresses limitations in conventional PCR protocols by significantly reducing current PCR run times. This was achieved by developing a protocol that combines a low temperature differential (∆T) with short dwelling times. These modifications are complemented by adjusted primer and probe design parameters, alongside the use of basic yet efficient buffers. These combined innovations streamline qPCR reactions by significantly reducing denaturation and polymerisation times at lower denaturation temperatures (79–80 °C) and higher polymerisation temperatures (71–72 °C).

Moreover, the buffers enable effective reverse transcription, simplifying one-step RT-qPCR assays. This comprehensive approach is versatile, applicable across diverse pathogens and cellular mRNA targets, leading to the development of a universally adaptable technique we have called “FlashPCR”. This method completes PCR reactions within 10–15 min on standard qPCR cyclers, presenting significant promise for accelerating research and diagnostic applications on the next generation of faster instruments.

## 2. Results

### 2.1. Amplification across Denaturation and Polymerisation Gradients

Following the establishment of optimal probe, primer, and Taq polymerase concentrations (Appendix A), more than 50 component variations in a basic qPCR reaction buffer were evaluated to identify buffers that would enable two modifications to the conventional PCR protocol: (i) lower denaturation and higher annealing/polymerisation temperatures than conventionally considered practical and (ii) minimal ∆T between the two cycling temperatures whilst maintaining the reactions’ specificity, sensitivity, and repeatability. This was done using numerous denaturation and polymerisation gradients, a range of Taq polymerases, as well as different primer and probe designs. For the amplification of SARS-CoV-2, this resulted in a final selection of MyTaq non-hot start polymerase, a choice of buffers (B47 and B50) that recorded essentially the same results targeting the CoV-E assay, which has a G/C content of 44%. Figure 1A shows representative amplification plots and Cq values with B47 compared to two commercial master mixes (Com 1 and Com 2) on a 79 °C to 85 °C denaturation gradient (Protocol P2). This resulted in ∆Cqs (79 °C vs. 85 °C) of 0.83, 2.17 and 2.87 for B47, Com 1, and Com 2, respectively. Figure 1B shows representative amplification plots and Cq values from a 67 °C and 72 °C polymerisation gradient (Protocol P3) with ∆Cqs (72 °C vs. 67 °C) of 0.54, 1.81 and 3.40, respectively. These results were highly reproducible, regardless of whether cDNA or PCR amplicons were used as target DNA. This is demonstrated by the bar plots in Figure 1C,D obtained using B47 or B50 that show the ∆Cq values (±95% confidence intervals) from individual denaturation gradients (*n* = 65, of which 29 were PCR amplicons) and polymerisation gradients (*n* = 62, of which 26 were PCR amplicons) against the respective 85 °C denaturation or 67 °C polymerisation controls. These results indicate that an amplicon size of around 85 bp, primers, and a probe with Tms of around 71 °C and 75 °C, respectively, permitted effective amplification using a denaturation temperature of around 79 °C to 80 °C combined with a polymerisation temperature of around 70–71 °C.

### 2.2. Amplification Using Single Denaturation and Polymerisation Temperatures

Since a temperature gradient might not accurately reflect an assay’s performance under single qPCR cycling temperatures, four protocols with different ∆T values (P4–P7) were used to amplify SARS-CoV-2 cDNA with B47 and the CoV-E assay. Results were compared to the Cqs recorded using the standard protocol (P1). P1 and P7 were repeated five times, P4 and P5 four times, and P6 was run once. An additional repeat for P7 was included to increase our confidence in the repeatability of this, the most extreme protocol. The single run using P6 was due to an accidental mistyping of the polymerisation temperature. Results as assessed by ∆Cq values were comparable across all runs, with the average ∆Cq values (±95% CI) relative to the standard protocol (P1) being −0.03 (−0.44, 0.37) for the ∆T = 10 °C protocol (P4), −0.19 (−0.49, 0.1) for the ∆T = 9 °C protocol (P5), 0.49 (−0.13, 1.1) for the ∆T = 9 °C with a denaturation temperature of 79 °C (P6) and 0.56 (0.26, 0.86) for the ∆T = 8 °C protocol (P7) (Appendix A). The reduction in ∆T resulted in a reduction in run times from 31 min to 19 min.

The experiment was repeated on the BMS Mic qPCR instrument with modified protocols P7 and P8 and buffer 19 to account for instrument-specific variability. Results were similar, although the optimal polymerisation temperature was 69 °C and holding times were a little longer (Appendix A). On this instrument, run times were reduced from 33 min with P8 to 16 min with P9.

### 2.3. Amplification with Short Primers Modified with Pentabases

The purpose of these experiments was to determine whether the incorporation of modified bases into primers would allow the use of shorter primers whilst maintaining the ability to anneal at higher polymerisation temperatures. First, the performance of the standard 31-mer DNA CoV-E primers was compared to that of the 24- and 26-mer Pentabase primers PB-F and PB-R on denaturation (P10) and polymerisation (P3) gradients. Cq values across the denaturation gradient were similar for both primer sets, with ∆Cqs between 85 °C and 79 °C also comparable at 0.03 and 0.09, respectively, for DNA and PB primers (Appendix A). A similar result was obtained across the polymerisation gradient, with ∆Cqs of 0.7 and 0.53, respectively (Appendix A).

When the primers were compared using single denaturation and polymerisation temperatures, both sets of primers recorded similar ∆Cq values relative to the control P1 protocol (Appendix A). With protocol P5, the DNA and PB primers recorded ∆Cqs against protocol P1 of 0.025 (−0.35, 0.4) and −0.32 (−0.87, 0.22), respectively. With protocol P7, the respective ∆Cq values were 0.84 (0.1, 1.57) and 0.45 (−0.47, 1.36), indicating a marginally better performance of the Pentabase primers.

### 2.4. Limit of Detection (LoD) and PCR Efficiency

Absolute copy numbers of SARS-CoV-2 cDNA were determined using ddPCR and assay CoV-E, with the highest dilution containing 58 (range 52–70) copies/reaction (Figure 2A). The LoD was determined on the BioRad CFX by preparing two-fold dilutions of that sample to 29, 15, and 7 copies/reaction and subjecting them to 15 replicate qPCR reactions using B47 and protocol P11 with a 72 °C polymerisation temperature (Figure 2B). All reactions containing 58 copies recorded Cq values, whereas only 87%, 67%, and 33%, respectively, of the higher dilutions did so, which is indicative of a 95% LoD of around 45 copies (Figure 2C). The experiment was repeated with fresh dilutions of 198, 29, and 10 copies (Figure 2D), but using protocol P5 with a 71 °C polymerisation temperature (Figure 2E). This time all 18 dilutions of the three dilutions recorded Cq values. Since LoD is the measurand concentration that produces at least 95% positive replicates, the LoD working at 95% confidence is <10 copies.

The linearity and efficiency of the reaction were determined by running serial dilutions of SARS-CoV-2 PCR amplicons with assay CoV-E using protocol P1 with SensiFast master mix (Appendix A). The second run used protocol P5 and B47 with MyTaq polymerase (Appendix A). The Cqs recorded by the amplification plots resulted in standard curves indicative of a PCR efficiency of around 100% for both buffers (Appendix A).

### 2.5. Other Pathogens

The wider application of the low ∆T modified protocols was demonstrated by the successful amplification of the genomic DNA from a range of common pathogens. This required the use of alternative buffers, since B47 worked well on the polymerisation gradient, but performed less well on the denaturation gradient. This was most likely due to the G/C content of the *S. aureus* amplicon, which is much higher at 55% than the CoV-E amplicon. Figure 3 shows amplification plots recorded on denaturation (Figure 3A) and polymerisation (Figure 3B) gradients for the *S. aureus* assay amplified using protocols P12 and P13 and buffer 25 (B25). The ∆Cq between 90 °C and the lowest denaturation temperature of 80 °C was 0.02. The ∆Cq between 67 °C and the highest polymerisation temperature of 72 °C was −0.22.

A comparison of amplification results using two single temperature protocols P5 and P14 confirmed that amplification using a low denaturation, high polymerisation temperature and small ∆T protocol was comparable to a conventional qPCR run (Figure 3C).

gDNA from three further pathogens, *Candida auris* (49% G/C), *Aspergillus fumigatus* (42% G/C) and *Acanthamoeba castellanii* (50% G/C), was amplified with a slightly modified buffer, B27, on denaturation and polymerisation gradients using protocols P12 and P13, respectively. All three DNA samples were amplified with approximately equal efficiency across both gradients (Appendix A).

### 2.6. RT and PCR

The practicability of using our basic buffers in reverse transcription reactions was assessed by reverse-transcribing human breast cancer mRNA with EpiScript (ES), SuperScript IV (SS) or UltraScript (US-2) using either their respective native buffers supplied by the manufacturers or B47. Aliquots of the resulting cDNAs were amplified using protocol R1 with PCRBio SyGreen master mix using primers targeting GAPDH (G/C content 57%). Melt curve analysis showed single, identical melt curves for each of the amplicons and similar Cq values (Appendix A). A comparison of the Cq values recorded by each of the RTases in the native buffers supplied by the manufacturers and B47 showed that the RTases performed equally well in B47, with ∆Cqs of −0.05 (−0.81, 0.71) (ES), 0.47 (−0.15, 1.1) (SSIV), and −0.67 (−1.63, 0.3) (US-2) (Appendix A).

The same cDNA samples were amplified in a dual-plex reaction using assays targeting TSG-6 (49% G/C) and HGF-1 (46% G/C) assays with protocol R1 and NEB Luna probe master mix. Amplification products were detected using FAM and HEX hydrolysis probes, respectively. As with GAPDH, amplification plots and Cq values were similar, regardless of which buffer was used for the RT step (Appendix A). There was little buffer-dependent difference in ∆Cq values between samples for either TSG-6 at 0.82 (−0.44, 2.08) (ES), 0.48 (−0.50, 1.47) (SS) and 0.52 (−1.14, 0.11) (US-2) or HGF-1 (0.44 (0.12, 0.76) (ES), 0.49 (−0.31, 1.28) (SS) and 0.61 (−1.21, −0.01) (US-2) (Appendix A).

The repeatability of these results was assessed by using EpiScript or UltraScript-2 to reverse-transcribe two different samples of SARS-CoV-2 gRNA targeting CoV-E in either native buffer or B47, followed by amplification with SensiFast master mix using protocol R1. The Cqs recorded using EpiScript and run on a Techne PrimePro 48 were the same regardless of whether native buffer (27.55 ± 0.22) or B47 (27.56 ± 0.20) was used for the RT step (Appendix A), resulting in a ∆Cq value of 0.015 (95% CI: −0.16, 0.19) (Appendix A). A repeat experiment using Ultrascript-2 in its own buffer (32.85 ± 0.39) or B47 (33.24 ± 0.8) and amplified with B47/MyTaq on the BioRad Opus using protocol R2 gave similar results (Appendix A) with the ∆C value being 0.4 (0.17, 0.62) (Appendix A).

Many RT-qPCR reactions use a one-step format and since B47 was an efficient buffer for RTs, we assessed the ability of B47 to act as a combined RT and qPCR buffer. One-step RT-qPCR reactions were set up with SARS-CoV-2 gRNA targeting either the CoV-E assay or a previously described assay targeting the Nsp10 gene (45% G/C) [27]. The reactions were carried out with PCRBio’s Clara, NEB’s Luna or B47/UltraScript (US)/MyTaq master mixes using protocol R1 on the Techne Prime Pro 48 instrument. All 1-step RT-qPCR reactions worked equally well and recorded similar results for either target (Figure 4A,B). To test the compatibility with other buffers and assays, human breast cancer mRNA samples were reverse-transcribed with ES in four buffers, B25, B27, B47 or B50, targeting TSG-6 and HGF-1 in a 1-step RT-qPCR dual-plex assay on a BioRad CFX. B47 and B50 recorded the lowest Cq values, with B25 close behind and B27 working less well (Figure 4C,D).

The experiment was repeated, except that this time ES was also combined with two alternative Taq polymerases, GoTaq (Promega) and ExTaq (Takara). One-step RT-qPCR assays were carried out on both the PrimePro 48 (Appendix A) and the BioRad CFX thermal cyclers (Appendix A) using protocol R2. The B47-based master mixes gave broadly similar results for TSG-6 (Appendix A) and HGF-1 (Appendix A), regardless of which polymerase or instrument was used. Interestingly, the B47 1-step reagent recorded, if anything, slightly lower Cqs than the two commercial master mixes used for comparison.

We have previously reported that a 1-step RT-qPCR reaction can be carried out without a dedicated RT-step [28]. This involves setting up the reactions, pipetting the samples onto microtitre plates, and spinning them at room temperature. The five minutes or so it takes to complete these steps was sufficient to complete the RT reactions. To see whether this approach worked with our B47 1-step reagent, breast cancer-derived mRNA samples were subjected to 1-step RT-qPCR reaction using conventional protocol R2 (Appendix A), PCRBio’s Clara, NEB’s Luna or B47 1-step RT-qPCR master mixes and the duplex TSG-6/HGF-1 assay. Reactions were carried out on the BioRad CFX Connect. The B47 master mix recorded lower Cq values than Clara (*p* < 0.002) and Luna (*p* < 0.02) for TSG-6, whereas Cq values were similar for HGF-1 (Appendix A). When the experiment was repeated using protocol R3 (Appendix A), the B47/ES/MyTaq buffer recorded significantly lower Cqs for both TSG-6 and HGF-1 (Appendix A). A comparison of Cq values between assays carried out without and with a dedicated RT step revealed ∆Cq for TSG-6 and HGF-1 of 1.6 (1.45, 1.75) and 3.1 (2.83, 3.41) for PCRBio’s Clara, 1.9 (1.41, 2.36) and 4.2 (3.96, 4.33) for NERB’s Luna and 0.3 (0.23, 0.35) and 1.3 (1.13, 1.52) for B47/ES/MyTaq, respectively (Appendix A).

The repeatability of these results was assessed using assays targeting TSG-6, HGF-1, GAPDH or CDKN1A (55% G/C) with human fibroblast mRNA and the B47/ES/MyTaq 1-step master mix. Runs were carried out in parallel using protocols R2 or R3, with 16 replicates on the BioRad CFX and 9 replicates on the Opus instruments. Amplification plots recorded using protocol R2 on the CFX (Figure 5A) and Opus (Figure 5B) or protocol R3 on the CFX (Figure 5C) and Opus (Figure 5D). The ∆Cq values confirmed that the lack of a dedicated RT step did not affect the performance of the assays (Figure 5E).

Finally, 1-step RT(−) reactions were set up in duplicate without dedicated RT steps with 4 replicates run using four different cycling conditions (R4 to R7) on the BioRad CFX Connect. The results shown in Figure 6 demonstrate that the combination of no dedicated RT step and low ∆T qPCR is a feasible alternative to conventional 1-step RT-qPCR protocols.

## 3. Discussion

This study streamlines and accelerates qPCR applications by introducing several key changes that extend the potential applications of qPCR-based technologies in molecular diagnostic settings:Primer designs that incorporate Tms higher than conventionally recommended;Modifications such as Pentabases that allow the use of short primers whilst maintaining a high Tm;Flexible probe designs that tolerate overlap of the 3′-ends of probes with 3′-ends of primers binding to opposite strands;Simple buffers that balance the requirement to denature PCR amplicons with the ability of primers to hybridise and prime polymerisation;Low ∆T protocols that involve running qPCR reactions at denaturation temperatures of approximately 80 °C and polymerization temperatures around 70 °C;Short cycling times that minimise qPCR run times;Wide applicability demonstrated through the targeting of genomic DNA from various common pathogens;Efficient one- and two-step RT-qPCR amplification of viral gRNA and cellular mRNA, even in the absence of a dedicated RT step.

These modifications, resulting in a protocol we have called “FlashPCR” are easy to implement, and it is likely that they can be refined even further. One of the assessments that define an assay’s “analytical sensitivity” is the LoD [29]. It is an important metric that delineates the smallest quantity or concentration of a target that can be detected consistently by a qPCR assay [30]. The LoD achieved in this study, approximately 10 copies per reaction, positions the technique among the most sensitive detection methods available. This level of sensitivity is important, particularly for early disease detection and for monitoring low-level pathogen presence, although it is important to remember that diagnostic sensitivity is dependent on more than a low LoD.

The study’s emphasis on maintaining PCR efficiency under the established low ∆T protocol, as compared to standard conditions, underscores the reliability and accuracy of the newly developed approach. The modified protocol provides reassurance that the gains achieved in terms of time and energy efficiency are not attained at the expense of the fundamental performance metrics of the qPCR reaction. A critical aspect of this innovation lies in its compatibility with a diverse array of Taq polymerases. This means that there will be no dearth of reagent companies able to provide modified master mixes, thus helping to keep costs down. The ability to detect a wide spectrum of pathogens, ranging from viruses to protozoa, is imperative in diagnostics, where pathogen diversity demands versatile detection techniques. The demonstrated versatility of the approach lends itself to a broad range of clinical applications, allowing for a unified methodology to detect various diseases. Comparable benefits extend to food safety, veterinary, and agricultural diagnostics.

Additionally, this study addresses the reverse transcription (RT) step, widely used to analyse gene expression pattern, screen for RMA biomarkers, and play a pivotal role in the detection of RNA-based pathogens [31]. The compatibility of the optimised buffers with the RT step, facilitating a one-step reaction, streamlines the diagnostic process. This flexibility not only simplifies the workflow but also minimises the chances of contamination and reduces the potential for errors, making the method even more robust and suitable for routine diagnostics.

Tailoring primers and probes to accommodate FlashPCR’s more stringent conditions requires deviations from established design conventions. Nevertheless, integrating these modifications into primer and probe design does not present significant challenges and does not demand a more exhaustive assessment of primer-probe combinations than typically undertaken. Consequently, the adaptation of existing assays and the creation of novel ones need not pose substantial hurdles. Notably, despite the higher Tm, crucial for our enhanced PCR conditions, these adjustments have not hampered the priming process during reverse transcription for cDNA synthesis. This underscores the feasibility of implementing altered design guidelines without compromising primer–probe functionality, facilitating a smooth transition to assays compatible with our optimised PCR parameters.

Our results also indicate that the use of Pentabase primers, which are characterised by their unique five-membered heterocyclic structure, offers distinct advantages. Their enhanced binding affinity for targeting sequences reduces nonspecific amplification and helps with accurate quantification. It also allows for the use of shorter primers, which increases the flexibility of assay location and permits the design of shorter PCR amplicons. This facilitates faster amplification cycles and can also reduce the probability of secondary structures that might hinder primer binding and subsequent amplification [32].

Whilst our modifications significantly reduce the duration of qPCR runs on standard instruments, an important future advantage will be increased speed coupled with lower-specification instrument design requirements, which will facilitate the implementation of PoC procedures [33]. This has enormous implications, as it is obvious that there is an increased need for early, quick and decentralised diagnostic testing [34], especially in resource-limited settings [35]. The ability to expedite the qPCR process without compromising sensitivity or accuracy [36] makes this approach a prime candidate for integration with emerging rapid diagnostics solutions. The substantial reduction of processing time, whilst maintain the performance characteristics of conventional protocols, will enable timely decision-making by healthcare professionals, facilitate prompt treatment initiation, and help implement efficient infection control measures. Moreover, incorporation into PoC procedures will enhance the responsiveness of public health agencies in managing outbreaks and instituting preventive measures, potentially limiting disease transmission [37,38].

Our results were achieved using standard qPCR plates running notably smaller volumes than conventionally employed. This deviation from standard volume usage shows that these current tools might be suboptimal for maximizing the potential of the modifications reported here. It stands to reason that employing dedicated thin materials with enhanced thermal properties, precisely tailored to optimal volume sizes for the wells, could generate even better results. Such modifications would also enhance uniformity across wells, ensuring more consistent temperature profiles during the PCR process. This optimisation in vessel design, coupled with precise volume considerations, holds promise for further improvement in the performance of the buffers in amplifying PCR amplicons, warranting exploration in future investigations to refine experimental conditions for enhanced assay robustness and reproducibility.

Finally, we evaluated numerous buffers, incorporating various combinations and concentrations of 1,2 propanediol, 1,3 propanediol, ethylene glycol, DTT, trehalose, betaine, DMSO, Triton-X-100, Tween 20, Nonidet P40 and formamide, alongside different concentrations of other components such as (NH_4_) _2_SO_4_, MgCl_2_ and KCl. This resulted in the selection of five buffers with only slight compositional variations that enabled the successful amplification of a range of PCR amplicons characterised by varying G + C content. We observed no discernible pattern dictating the superior performance of one buffer over another. Our findings indicate that, whilst the efficacy of buffers in supporting PCR amplification is correlated with the G + C content of the amplicons [39], this may not be the most important determinant. It prompts speculation regarding additional factors influencing the compatibility of individual assays with specific buffer compositions. Among these potential considerations is the variability inherent to different PCR instruments, which might necessitate subtle adjustments in buffer composition for optimal performance. Such instrument-specific variability, including variations in thermal cycling mechanisms or temperature uniformity, could feasibly influence the compatibility of individual assays with specific buffer formulations. Other variables, such as secondary structure complexities, primer-template interactions, or even the unique sequence context within the target region, likely contribute to a buffer’s impact on amplification efficiency. It is clear that our experiments serve as preliminary investigations, laying the groundwork for reagent companies to use their expertise toward refining a universal buffer formulation.

## 4. Materials and Methods

### 4.1. Reagents and qPCR Instruments

The details of all commercial reagents, plasticware and instruments are listed in Table 1.

The recipes for buffers 19 (B19), 25 (B25), 27 (B27), 47 (B47) and 50 (B50) are shown in Table 2. qPCR reactions were carried out using four different qPCR instruments: 96-well cyclers CFX Connect and a CFX Opus (BioRad, Watford, UK), a 48-sample Mic magnetic induction instrument (Bio Molecular Systems, London, UK) or a 48-well Techne PrimePro 48 cycler (Cole Palmer, St. Neots, UK). The ddPCR runs were carried out on a BioRad QX200 instrument.

### 4.2. Primers and Probes

All assays were designed using the Beacon Designer qPCR assay design software package (V. 8.21, Premier Biosoft, San Francisco, CA, USA). Sequences specifying the E-gene from SARS-CoV-2, human TNFα-induced protein (TSG-6), hepatocyte growth factor 1 (HGF-1), glyceraldehyde 3-phosphate dehydrogenase (GAPDH), and cyclin-dependent kinase inhibitor 1 (CDKN1), as well as rRNA sequences from *Candida auris*, *Aspergillus fumigatus*, *Acanthamoeba castellanii* and *Staphylococcus aureus*, were downloaded from the NIH National Centre for Biotechnology Information website. Primers and probes were designed with manual adjustments aimed at obtaining short amplicons amplified by primers with high melting temperatures (Tm). Overlap of the 3′-end of probes and the 3′-end of primers hybridising to the reverse strand was permitted. The specificity of primers, probes, and amplicons was analysed in silico using Primer-BLAST (https://www.ncbi.nlm.nih.gov/tools/primer-blast/, last accessed on 23 February 2024) and BLAST (https://blast.ncbi.nlm.nih.gov/Blast.cgi/, last accessed on 23 February 2024). All oligonucleotides, except the Pentabase ones, were synthesised and lyophilised by Sigma Aldrich. Pentabases were synthesised by Pentabase AS (Odense, DK). Upon receipt, all were resuspended in sterile RNase-free water at 100 µM and stored in aliquots at −20 °C. Table 3 lists the details of the primers and probes together with their melting temperatures (Tms) and the lengths of the resulting PCR amplicons.

### 4.3. RNA Extractions

SARS-CoV-2 BA 5.1 genomic RNA (gRNA) was extracted from saliva samples using the Quick-RNA MiniPrep Plus Kit (Zymo Research, Irvine, CA, USA) using the method described for saliva and buccal cells, except that the RNA was eluted in 50 µL of RNase-free water. This RNA was assessed for inhibitors by using a 1-step RT-qPCR assay to compare the quantification cycles (Cqs) recorded by the amplification of neat RNA and a 1:10 dilution, but not for integrity, and stored at −80 °C.

Total human RNA was prepared from breast cancer biopsies using the RNeasy lipid tissue mini kit (Qiagen, Manchester, UK) or from primary fibroblast tissue culture cells using the RNeasy tissue mini kit (Qiagen) without any modifications. RNA samples were quantified and quality-assessed according to the MIQE guidelines [29] using a Nanodrop 2000 (ThermoFisher, Waltham, MA, USA) and their integrity was assessed using an Agilent Bioanalyser 2100 (Agilent, Stockporyt, UK). All RNAs recorded RIN values of above 9 and were stored at −80 °C. Whilst it is acknowledged that RIN values may not accurately reflect the integrity of mRNA, it does provide some measure of confidence with regards to RNA quality and remains a useful indicator until a better standard is developed.

### 4.4. cDNA Synthesis

#### 4.4.1. SARS-CoV-2

cDNA was synthesised from SARS-CoV-2 gRNA using SuperScript IV reverse transcriptase (SSIV, ThermoFisher, Waltham, MA, USA) and buffer. All reagents were kept on ice prior to carrying out the RT step in 40 µL using 100 U RT, 100 ng random primers and 0.2 mM of each dNTP. Reaction conditions were 5 min at 25 °C, 5 min at 55 °C and 5 min at 85 °C. cDNA samples were diluted with 40 µL of RNase-free water and stored at −20 °C.

#### 4.4.2. Human RNA

Human breast cancer- or fibroblast-derived total RNA samples were reverse-transcribed with UltraScript (US, PCRBio, London, UK), UltraScript 2 (US2, PCRBio), SuperScript (SS4, ThermoFisher, Waltham, MA, USA) or EpiScript (ES, Biosearch, Petaluma, CA, USA), either in the RT buffers supplied with the kits or one of the buffers listed in Table 2. All reagents were kept on ice prior to carrying out the RT step in 20 µL using 100 U RT, 100 ng random primers and 0.2 mM of each dNTP. Reaction conditions were 25 °C for 5 min, 50 °C (55 °C for SSIV) for 5 min, and 85 °C for 5 min. cDNA samples were diluted with equal volumes (20 µL) of nuclease-free water (ThermoFisher, Waltham, MA, USA) and stored at −20 °C.

### 4.5. qPCR Reactions

Unless otherwise stated, 2.5 µL reaction volumes were used. cDNA (0.25 µL per reaction) was used for most experiments except for the standard curves and LoD experiments. These reactions were run using dilutions of PCR amplicons. All reactions were set up at room temperature using pre-cooled reagents using one of the protocols (P1–15) described in Table 4. Each experiment was carried out using premixes containing all reagents except the one being tested (i.e., enzyme, cDNA, buffer). For reactions carried out using the BioRad or Cole Palmer instruments, concentrations of *Taq* polymerase, primers, and probes were determined empirically using protocol P1 and used at 0.06 U/2.5 µL reaction, 0.5 µM and 0.2 µM final concentrations, respectively. On the Mic, *Taq* polymerase, primer, and probe concentrations were 0.25 U/2.5 µL, 1 µM, and 0.4 µM, respectively. Depending on the qPCR instrument, assays were carried out in heat-sealed white qPCR 96 well plates (BioRad), adhesive-sealed well plates (Cole Palmer) or 4-tube strips (BMS, London, UK).

### 4.6. ddPCR Reactions

The BioRad protocol was followed exactly for generating droplets (manual 186–4002) as well as setting up, running, and analysing ddPCR runs (manual 1864001/3). Reactions were carried out using ddPCR Supermix for probes (BioRad), with primers and probes at concentrations of 0.9 µM and 0.25 µM, respectively, in 20 µL volumes. When mixed with the droplet generating oil (BioRad), this resulted in final ddPCR volumes of 40 µL produced on the QX200 droplet generator (BioRad). Thermal cycling was performed on a C1000 thermal cycler (BioRad) with a ramp rate of 2 °C using the following thermal cycling protocol recommended by BioRad: 95 °C 10 min, followed by 40 cycles at 95 °C for 15 s and 60 °C for 60 s, and a final 98 °C 10 min step. Droplets were counted using the QX200 droplet reader (BioRad).

### 4.7. Pathogen DNA

*Aspergillus fumigatus* DNA was extracted from fungal cultures, as described previously [40]. *Acanthamoeba castelanii*, *Candida auris*, and *Staphylococcus aureus* DNA was purchased from Vircell Microbiologists (Table 1).

### 4.8. 1-Step RT-qPCR Reactions

One-step reactions were carried out with SARS-CoV-2 BA 5.1 gRNA, as well as human breast cancer or fibroblast derived total RNA. PCRBio’s Clara or NEB’s Luna 1-step master mixes were included as positive reagent controls. B25, B27, B47 or B50 were used with 100 U of US or ES RTs, 0.5 µM/0.25 µM primer/probe mix, and 0.06 U/2.5 µL of MyTaq, GoTaq (Promega, Madison, WI, USA) or ExTaq (Takara, Saint-Germain-en-Laye, F). Reactions were carried out using one of two alternative RT steps:

A. All plasticware and reagents were kept on ice whilst dispensing the individual reaction components. Plates were sealed, spun for 5–15 s at 4 °C in a refrigerated centrifuge, placed in a qPCR thermal cycler and assays were run using protocols R1 or R2 (Table 4).

B. All plasticware and reagents except the RTs were kept at room temperature and reactions were set up at room temperature. Plates were sealed and spun for 5–15 s at room temperature before being placed in a qPCR thermal cycler. This process took around 5 min, depending on the number of samples being analysed. Assays were then run using one of the protocols R3-R7 (Table 4).

### 4.9. Data Analysis

All qPCR data were imported and analysed in Microsoft Excel for Mac v.16.80 and PRISM for Mac v.10. ddPCR results were analysed and exported using QX Manager v.2.1 software. ∆Cq values are shown with ±95% Confidence Intervals (CI) rather than ±Standard Deviation (SD). The resulting wider interval acknowledges the level of uncertainty in the measurement and provides a more transparent representation of the potential range of ∆Cq values.

## 5. Conclusions

This study introduces a ground-breaking advancement in the way PCR is carried out. The optimisation of buffers and assay redesign to enable qPCR under low ∆T conditions offers a host of benefits, from faster reaction times and reduced energy consumption to broader application potential. The compatibility with diverse Taq polymerases, the inclusion of the RT step, and the notable LoD and PCR efficiency collectively highlight its potential for point-of-care diagnostics. Its impact spans diverse fields, including clinical diagnostics, epidemiology, environmental monitoring, and personalised medicine, paving the way for significant advancements in disease detection and patient care.

## 6. Note Added in Proof

We have recently had the opportunity to run FlashPCR assays on a QiaQuant 96 qPCR instrument (Qiagen, Manchester, UK), with the only limitation being that the instrument requires a minimum polymerisation time of 3 s to enable scanning of the 96-well plate. Compared with a standard 1 s at 95°C initial denaturation, followed by cycling for 5 s at 95 °C and 15 s at 60 °C, we achieved ∆Cq values of 0.56 ± 95% CI 0.42, 0.70 (1 s at 80 °C followed by cycling for 1 s at 80 °C and 3 s at 70 °C), −0.17 ±−0.04, −0.30 70 (1 s at 80 °C followed by cycling for 1 second at 80 °C and 3 s at 71 °C), 0.83 ± 95% CI 0.90, 0.76 (1 s at 80 °C followed by cycling for 1 s at 80 °C and 3 s at 72 °C) and 1.08 ± 0.81, 1.35 (1 s at 79 °C followed by cycling for 1 s at 79 °C and 3 s at 71 °C).

## Figures and Tables

**Figure 1 ijms-25-02773-f001:**
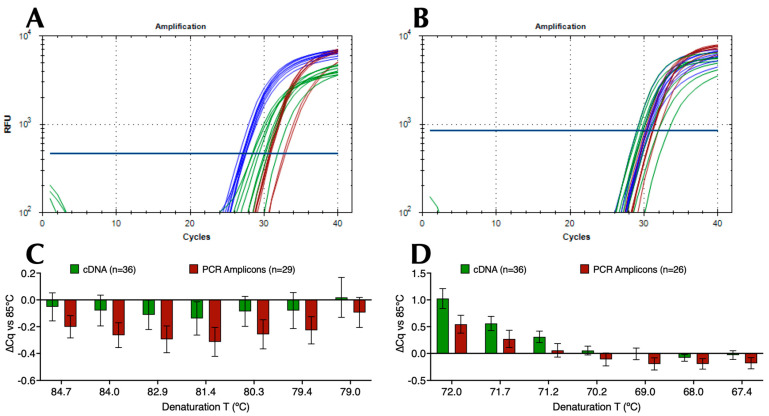
Amplification of SARS-CoV-2 cDNA with assay CoV-E. Horizontal lines show the position of the threshold used to calculate Cq values. (**A**) Amplification plots for B47 (blue), commercial master mixes 1 (brown) and 2 (green) targeting SARS-CoV-2 cDNA using denaturation protocol P2 on a BioRad CFX Connect. (**B**) Amplification plots recorded for B47 (blue), commercial master mixes 1 (brown) and 2 (green) targeting SARS-CoV-2 cDNA using polymerisation protocol P3 on a BioRad CFX Connect. (**C**) ∆Cq values (±95% CI) against 85 °C recorded at each denaturation temperature with B47 and B50 and cDNA (green, *n* = 36) or PCR amplicons (brown, *n* = 29) calculated from combining the data acquired on BioRad CFX Connect and Opus instruments. (**D**) ∆Cq values (±95% CI) against 67 °C at each polymerisation temperature with B47 or B50 and cDNA (green, *n* = 36) or PCR amplicons (brown, *n* = 26) calculated from combining the data acquired on BioRad CFX Connect and Opus instruments. All Cq values are listed in the Appendix A.

**Figure 2 ijms-25-02773-f002:**
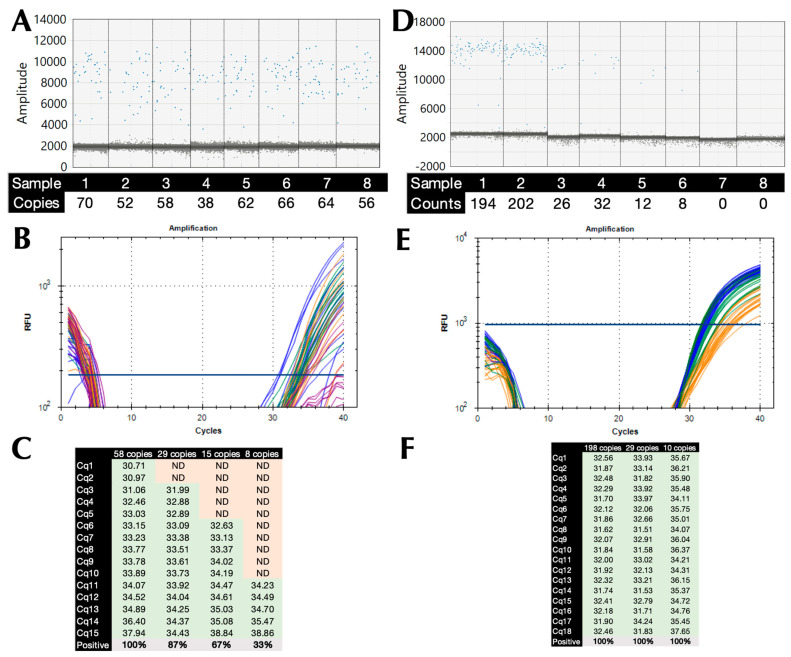
Limit of detection. (**A**) 1-D amplitude plot of 8 replicates of the highest dilution of the SARS-CoV-2 cDNA sample, with the positives (blue) clearly distinguished from the negatives (grey). The copy numbers/reaction were determined by the BioRad QX200 instrument software. (Version 2.1, BioRad, Watford, UK). (**B**) Amplification plots recorded for each of the replicate reactions (58 copies blue, 29 copies green, 15 copies orange, 8 copies red). Horizontal lines show the position of the threshold used to calculate Cq values. (**C.**) Cq values (green) recorded for each of the replicate reactions, with absence of amplification shaded red. (**D**) 1-D amplitude plot of duplicate dilutions of SARS-CoV-2 cDNA sample, including two NTCs, with the positives (blue) clearly distinguished from the negatives (grey). The copy numbers/reaction were determined by the BioRad QX200 instrument software (Version 2.1, BioRad, Watford, UK). (**E**) Amplification plots recorded for each of the replicate reactions (198 copies blue, 29 copies green, 10 copies orange). Horizontal lines show the position of the threshold used to calculate Cq values. (**F**) Cq values recorded for each of the replicate reactions. NTCs did not record Cq values.

**Figure 3 ijms-25-02773-f003:**
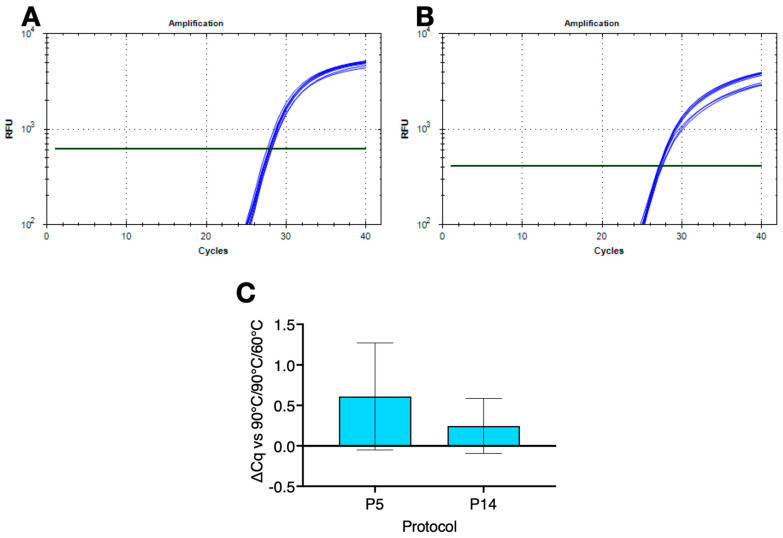
Amplification of S. aureus gDNA with B25 on denaturation or polymerisation gradients on a BioRad CFX Connect. (**A**) Amplification plots recorded on the denaturation gradient using protocol P12. Horizontal lines show the position of the threshold used to calculate Cq values. (**B**) Amplification plots on the polymerisation gradient using protocol P13. Horizontal lines show the position of the threshold used to calculate Cq values. (**C**) ∆Cq values (±95% CI) recorded with P5 (∆T = 9 °C) and P14 (∆T = 11 °C) versus protocol P1. All Cq values are listed in the Appendix A.

**Figure 4 ijms-25-02773-f004:**
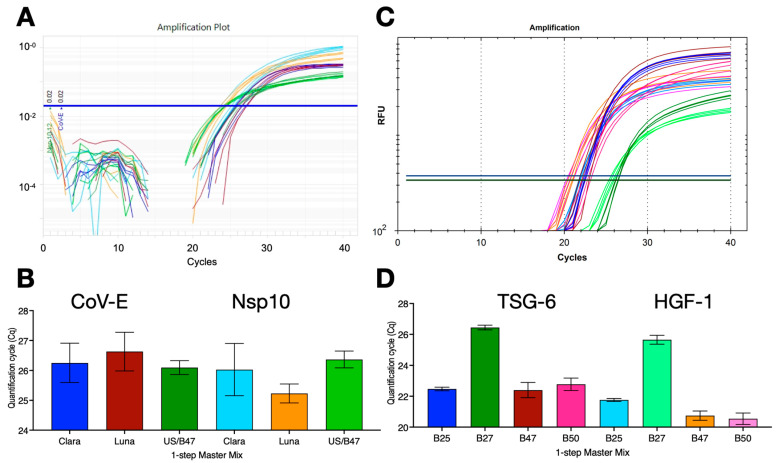
One-step RT-qPCR assays with SARS-CoV-2 genomic RNA or breast cancer mRNA. (**A**) Amplification plots recorded with CoV-E and Nsp10 assays and PCRBio Clara (dark blue/light blue), NEB Luna (brown/orange) 1-step RT-qPCR mastermixes, as well as the B47-based 1-step master mix with UltraScript RT and MyTaq polymerase (dark green/light green) run on a Hybaid PrimePro 48 qPCR instrument. Horizontal lines show the position of the threshold used to calculate Cq values. (**B**) Cq values ± SD. (**C**) Amplification plots recorded with dual-plex TSG-6 and HGF-1 assays and B25 (dark blue/light blue), B27 (dark green/light green), B47 (brown/orange) or B50 (dark pink/light pink)-based 1-step master mix with UltraScript RT and MyTaq polymerase BioRad CFX Opus qPCR instrument. Horizontal lines show the positions of the FAM and HEX thresholds used to calculate Cq values. (**D**) Plot of Cq values ± SD. All Cq values are listed in the Appendix A.

**Figure 5 ijms-25-02773-f005:**
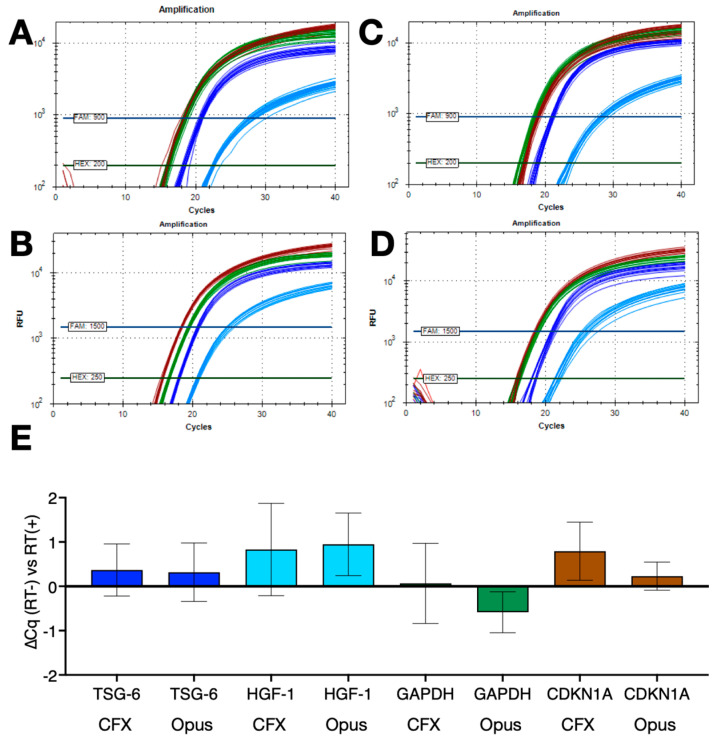
Repeatability of 1-step RT-qPCR protocols with and without dedicated RT steps and assays TSG-6 (blue), HGF-1 (light blue), GAPDH (green) or CDKN1 (brown). (**A**) Amplification plots recorded for 16 replicate reactions using RT+ protocol R2 on the BioRad CFX Connect. Horizontal lines show the positions of the FAM and HEX thresholds used to calculate Cq values. (**B**) Amplification plots recorded for 9 replicate reactions using RT+ protocol R2 on the BioRad Opus. Horizontal lines show the positions of the FAM and HEX thresholds used to calculate Cq values. (**C**) Amplification plots recorded for 16 replicate reactions using the no RT protocol R3 on the BioRad CFX Connect. Horizontal lines show the positions of the FAM and HEX thresholds used to calculate Cq values. (**D**) Amplification plots recorded for 9 replicate reactions using the no RT protocol R3 on the BioRad Opus. Horizontal lines show the positions of the FAM and HEX thresholds used to calculate Cq values. (**E**) ∆Cq values (±95% CI) of the reactions carried out with protocol R3 versus those carried out with protocol R2 (TSG-6 dark blue, HGF-1 light blue, GAPDH green, CDKN1A brown). All Cq values are listed in the Appendix A.

**Figure 6 ijms-25-02773-f006:**
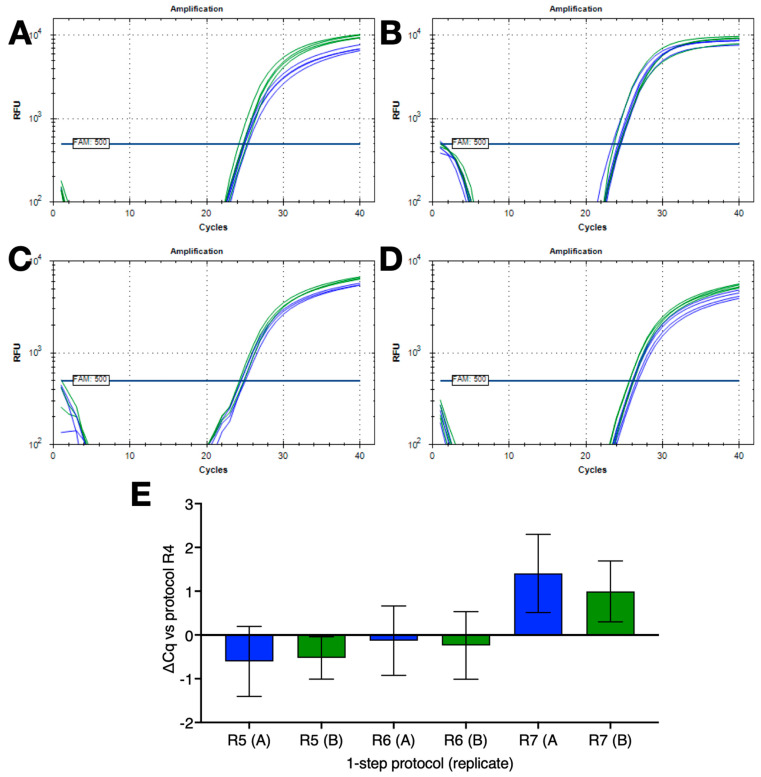
Comparison of 1-step RT-qPCR protocols with no dedicated RT steps coupled with FlashPCR protocol targeting SARS-CoV-2 gRNA. (**A**) Amplification plots recorded for replicate set-ups and reactions using protocol R4 (∆T = 30 °C) on a BioRad CFX instrument. Horizontal lines show the positions of the FAM and HEX thresholds used to calculate Cq values. (**B**) Amplification plots recorded for replicate set-ups and reactions using protocol R5 (∆T = 9 °C) on a BioRad CFX instrument. Horizontal lines show the positions of the FAM and HEX thresholds used to calculate Cq values. (**C**) Amplification plots recorded for replicate set-ups and reactions using protocol R6 (∆T = 10 °C) on a BioRad CFX instrument. Horizontal lines show the positions of the FAM and HEX thresholds used to calculate Cq values. (**D**) Amplification plots recorded for replicate set-ups and reactions using protocol R7 (∆T = 9 °C) on a BioRad CFX instrument. Horizontal lines show the positions of the FAM and HEX thresholds used to calculate Cq values. (**E**) ∆Cq values (±95% CI) of the various reactions versus those carried out with protocol R4 (∆T = 30 °C). All Cq values are listed in the Appendix A.

**Table 1 ijms-25-02773-t001:** Reagents and instruments.

Supplier	Reagent/Instrument/Software	Part No.
ABI, Warrington, UK	TaqMan Fast Advanced Master Mix	4444553
Agilent, Stockport, UK	2100 Bioanalyzer	G2939BA
Bioline, London, UK	SensiFast SYBR No-ROX	BIO-98050
Bioline, London, UK	MyTaq DNA polymerase	BIO-21105
Bioline, London, UK	SensiFast Probe No-ROX	BIO-86050
Biomolecular Systems, London, UK	Mic qPCR Cycler	N/A
Biomolecular Systems, London, UK	4-tube strip	MIC-tubes
BioRad, Watford, UK	CFX Connect	N/A
BioRad, Watford, UK	CFX Opus	N/A
BioRad, Watford, UK	ddPCR probe supermix (-dUTP)	186-3024
BioRad, Watford, UK	CFX QX200 Droplet Generator	186-4002
BioRad, Watford, UK	CFX QX200 Droplet Reader	186-4003
BioRad, Watford, UK	Droplet generation oil for probes	186-3005
BioRad, Watford, UK	PX1 PCR plate sealer	181-4000
BioRad, Watford, UK	qPCR heat seal	181-4030
BioRad, Watford, UK	ddPCR heat seal	181-4040
BioRad, Watford, UK	Skirted 96-well plates	HSP9645
BioRad, Watford, UK	CFX Maestro Software V. 2.3	12013758
BioRad, Watford, UK	QX Manager Software V. 2.1.	N/A
Biosearch, Petaluma, CA, USA	EpiScript RNase H- Reverse Transcriptase	ERT12925K-ENZ
Cole Palmer, St., Neots, UK	Techne Prime Pro 48 cycler	WZ-93945-14
Cole Palmer, St., Neots, UK	Techne plate seal	MB0481
Cole Palmer, St., Neots, UK	Techne 48 well plates	MB0482
IDT, Leuven, Belgium	PrimeTime Master Mix	1055772
Merck, Gillingham, UK	KAPA	KM4701
NEB, Hitchin, UK	LyoPrime Luna Probe One-Step RT-qPCR Mix	L4001SVIAL
PCRBio, London, UK	qPCRBIO Probe Blue Mix	PB20.27-05
PCRBio, London, UK	UltraScript 2	PB30.33
PCRBio, London, UK	UltraScript	PB30.12
PCRBio, London, UK	Clara Probe 1-Step Mix	PB25.83-01
Pentabase AS, Odense, DK	Pentabase primers	N/A
Premier Biosoft, San, Francisco, CA, USA	Beacon Designer 8.21	N/A
Promega, Southampton, UK	GoTaq Probe qPCR master mix	A610A
Promega, Southampton, UK	GoTaq DNA polymerase	M7845
Qiagen, Manchester, UK	RNeasy Mini Kit	74104
Qiagen, Manchester, UK	RNeasy Lipid Tissue Mini Kit	74804
Quanta, Beverly, MA, USA	PerfeCta qPCR Toughmix	84196
Quanta, Beverly, MA, USA	PerfeCta Multiplex qPCR Toughmix	84263
Sigma, Haverhill, Aldrich	Primers and probes	
Takara, Saint-Germain-en-Laye, F	Ex Taq probe premix	RR390
Takara, Saint-Germain-en-Laye, F	ExTaq DNA polymerase	RR001A
ThermoFisher, Waltham, MA, USA	Nanodrop spectrophotometer	N2000
ThermoFisher, Waltham, MA, USA	Nuclease-free water	AM9922
ThermoFisher, Waltham, MA, USA	SuperScript IV	18091200
Qiagen, Manchester, UK	QiaQuant 96	9003000
Qiagen, Manchester, UK	qPCR skirted plates	209002
Zymo, Research, Irvine,, CA, USA	Quick-RNA Miniprep Plus Kit	D7005

**Table 2 ijms-25-02773-t002:** Components for buffers 19, 25, 27, 47, and 50.

2× Buffer	B19	B25	B27	B47	B50
Tris pH 8.8 (mM)	20	20	20	20	20
KCl (mM)	100	100	100	100	100
MgCl_2_ (mM)	10	10	10	10	10
1,2 Propanediol (M)	0.5	0.5	0.4	1.0	1.0
1,3 Propanediol (M)	0.4	0.4	0.4	-	-
Ethylene Glycol (M)	0.6	0.6	0.6	-	0.16
Trehalose (M)	0.3	0.3	0.2	0.2	0.2
BSA (mg/mL)	0.2	0.2	0.2	0.2	0.2
dNTP (M)	0.4	0.4	0.4	0.4	0.4
Formamide (%)	0.5	-	-	-	-

**Table 3 ijms-25-02773-t003:** Details of targets, primers, probes, and amplicons. PB-F and PB-R are Pentabase primers. Capital letters indicate base substitution by an LNA.

Target	Accession No	Primers	Sequence (5′-3′)	Tm (°C)	Amplicon
**SARS-CoV-2**	NC_045512.2	CoV-EF	GTGGTATTCTTGCTAGTTACACTAGCCATCC	72.1	84bp
		CoV-ER	AAGACTCACGTTAACAATATTGCAGCAGTAC	71.2	
		PB-F	TCTTGCTAGTTACACTAGCCATCC	68.4	72bp
		PB-R	TCACGTTAACAATATTGCAGCAGTAC	68.2	
**TNFα induced protein**	NM_007115	TSG-6F	TCGCAACTTACAAGCAGCTA	65.8	85bp
		TSG-6R	CCAACTCTGCCCTTAGCC	66.1	
**Hepatocyte growth factor 1**	NM_0006501	HGF-1F	TCACAAGCAATCCAGAGGTAC	65.9	76bp
		HGF-1R	TTGCAGGTCATGCATTCAAC	65.5	
**GAPDH**	NM_002046	GAPDH-F	AGCCACATCGCTCAGACA	67.4	75bp
		GAPDH-R	TGACCAGGCGCCCAATAC	68.5	
**CDKN1A**	NM_000389	CDK-F	CTGGAGACTCTCAGGGTCGAA	68.9	98bp
		CDK-R	GGATTAGGGCTTCCTCTTGGA	67.3	
** *Staphylococcus aureus* **	OR365499.1	SA-F	GCGGTGAAATGCGCAGAGATATGGA	73.3	77bp
		SA-R	GCACATCAGCGTCAGTTACAGACCA	72.9	
** *Candida auris* **	OQ581784.1	CA-F	AACGGATCTCTTGGTTCTCGCATCG	72.7	70bp
		CA-R	CGTCTGCAAGTCATACTACGTATCGCAT	72.2	
** *Acanthamoeba castellanii* **	KT185626.1	Aca-F	GTCGATTGAACCTTACCATTTAGAGGAAGG	70.8	74bp
		Aca-R	GATCCCTCCGCAGGTTCACCTAC	72.6	
** *Aspergillus fumigatus* **	OR415310.1	Asp-F	TTCTTGGATTTGCTGAAGACTAACTACTGCG	70.9	85bp
		Asp-R	CGATCCCCTAACTTTCGTTCCCTGAT	70.3	
		**Probes**	Sequence (5′-3′)	Tm (°C)	Type
		CoV-E-Pr	FAM-cacAcaAtcGaaGcgCagTaag-Q	74.9	LNA
		PB-Pr	FAM-CACACAATCGAAGCGCAGTAAGGAT-Q	72.5	Pentabase
		TSG-6-Pr	FAM-tccAtcCagCagCacaga-Q	77.2	LNA
		HGF-1-Pr	HEX-cgaAgtCtgTgaCattcct-Q	70.9	LNA
		GAPDH-Pr	FAM-tccGttcGacTccGacct-Q	75	LNA
		CDK-Pr	FAM-atgCtgGtcTgcCgcc -Q	77.4	LNA
		SA-Pr	HEX-acaCcaGtgGcgAagGcga-Q	83.9	LNA
		CA-Pr	FAM-tcgCtgCgttCttCatCgat-Q	76.7	LNA
		Aca-Pr	FAM-aagTcgTaaCaaGgtCtccg	75.1	LNA
		Asp-Pr	FAM-acAtcCttGgcGaaTgcTttc-Q	74.2	LNA

**Table 4 ijms-25-02773-t004:** qPCR and 1-step RT-qPCR protocols. Room temperatures describe the usual laboratory temperature of between 20 °C and 21 °C.

qPCR Protocol	Reverse Transcription		Denaturation or Activation		Cycling			
					Denaturation		Polymerisation	
	Temp (°C)	Time (min)	Temp (°C)	Time (s)	Temp (°C)	Time (s)	Temp (°C)	Time (s)
P1	N/A	N/A	90	15	90	1	65	1
P2	N/A	N/A	85	15	79-85	1	70	1
P3	N/A	N/A	85	15	85	1	67–72	1
P4	N/A	N/A	80	1	80	1	70	1
P5	N/A	N/A	80	15	80	1	71	1
P6	N/A	N/A	79	15	79	1	70	1
P7	N/A	N/A	79	15	79	1	71	1
P8	N/A	N/A	95	300	95	5	65	10
P9	N/A	N/A	80	15	80	5	69	3
P10	N/A	N/A	85	15	79–85	1	71	1
P11	N/A	N/A	80	1	80	1	72	1
P12	N/A	N/A	90	1	80–90	1	71	1
P13	N/A	N/A	90	1	90	1	67–72	1
P14	N/A	N/A	82	15	82	1	71	1
P15	N/A	N/A	95	120	90	2	65	1
RT-qPCR Protocol								

R1	50	5	95	120	90	2	65	1
R2	50	5	95	60	95	1	60	1
R3	Room Temp	5	95	60	95	1	60	1
R4	Room Temp	5	90	15	90	1	60	1
R5	Room Temp	5	81	15	81	1	70	1
R6	Room Temp	5	80	15	80	1	70	1
R7	Room Temp	5	80	15	80	1	71	1

## Data Availability

Most Figures include the Cq data supporting the conclusions published in this article. The other data are included with the Appendix A.

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
