# Peer review of "FlashPCR: Revolutionising qPCR by Accelerating Amplification through Low ∆T Protocols"

_ijms, 2024, doi:10.3390/ijms25052773_

Round 1
Reviewer 1 Report
Comments and Suggestions for Authors
1. In line 108 and line 109, the paper said that 'On this instrument run times were reduced from 33 minutes with P7 to 17 minutes with P8'. However, it seemed the Protocol 8 needed more time.
2. Did Figure2(F) has a negative control?
3. In line 453, the paper said 2.5 μL reaction volumes were used. The volumn is too small, can it work normally in 96 well plates(BioRad CFX) ? Please use an ordinary volume and 2.5 μL to compare.
4. What were the template volume and reaction volume in LoD experiments?
5. In Table 3. most of the denaturation and polymerisation times were 1 sec, it wasn't enough for BioRad CFX to collect fluorescence signals. How did you get the curves.
6. Chapter 4 'Materials and Methods' should be move to the front of results.
7. Too many tests were done, but there was no clear explanations of what the purposes of the tests were and why the tests were done like this. The methods, results and discussions were not integrated well.
8. Why did you use so many PCR machines? How to campare the results?
Author Response
In line 108 and line 109, the paper said that 'On this instrument run times were reduced from 33 minutes with P7 to 17 minutes with P8'. However, it seemed the Protocol 8 needed more time.
Apologies, these are typos overlooked because of the many protocols involved. As shown in Figure S2C, the protocols should have been labelled P8 and P9.
2. Did Figure2(F) has a negative control?
The sentence “NTCs did not record Cq values.” Was added to the Figure legend.
3. In line 453, the paper said 2.5 μL reaction volumes were used. The volumn is too small, can it work normally in 96 well plates(BioRad CFX) ? Please use an ordinary volume and 2.5 μL to compare.
A 2.5µL volume is not too small for a 96 well plate. We regularly run such small volumes and 1µL volumes on the PrimePro 48. The Figure below shows the amplification plots from 96 5µL reactions run on a CFX Connect (A), compared with 96 2.5µL reactions (B). It also shows the results recorded using 1µL volumes on the PrimePro 48 (C). It is evident that such small volumes result in highly reproducible Cq values.
4. What were the template volume and reaction volume in LoD experiments?
The conditions are those stipulated by BioRad in their manual, as described in section 4.6 ddPCR reactions. 20µL reagent volumes resulted in 40µL reaction volumes after the addition of oil.
In Table 3. most of the denaturation and polymerisation times were 1 sec, it wasn't enough for BioRad CFX to collect fluorescence signals. How did you get the curves.
We assume you are unfamiliar with this instrument, as the fluorescence reading step is carried out after each cycle. Hence the actual time at the annealing/polymerisation step is more than 1 second, allowing the fluorescence reading to take place. You can see the protocol as displayed by the instrument in Figures 2B and F.
6. Chapter 4 'Materials and Methods' should be move to the front of results.
This is the journal’s style, hence we cannot do this.
7. Too many tests were done, but there was no clear explanations of what the purposes of the tests were and why the tests were done like this. The methods, results and discussions were not integrated well.
We respectfully disagree, as seemingly do reviewers 2 and 3. The purpose of these tests was to develop a fast PCR protocol that is universally applicable and makes use of the latest improvements to enzymes and instruments. We have advocated faster PCR for many years and this paper provides all the detailed information required for anyowe to reproduce our results and apply our findings to their own research or diagnostic tests. These experiments were carried out to convince what we think will be a sceptical audience that this new PCR protocol/ assay/buffer combination is robust and reproducible. The discussion could have been massively extended, but this would have made the paper too dense. We think that the results have been clearly discussed and appropriate inferences were drawn.
8. Why did you use so many PCR machines? How to campare the results?
We used a variety of instruments to demonstrate the universality of our new PCR approach. Had we limited ourselves to a single instrument, the criticism would have been (rightly) that this is a method that may be specific to a single instrument.
Reviewer 2 Report
Comments and Suggestions for Authors
In the manuscript entitle: “FlashPCR: Revolutionising qPCR by Accelerating Amplification through low ΔT protocols”, the authors addressing a developed approach called FlashPCR to accelerate qPCR applications with short cycling times through optimization of buffers, primer/probe designs with different low ΔT conditions. The overall manuscript is well written and well- addressed. I have some comments:
Line 14: Abstract: If possible, I suggest providing the most important results in abstract to address which is the best condition of qPCR that defines the “FlashPCR” approach.
Line 99: the sentence “P1 and P7 were repeated five times, P4 and P5 four times and P6 was run once”. please provide the reason for that.
Line 106: “modified protocols P7 and P8 and buffer 19 to account for instrument-specific variability”, please provide the modification that has been made in M & M.
Line115: the sentence “the 24- and 26-mer Pentabase primers PF-F and PB-R on denaturation” what do you mean PF-F? is that PB-F?
Line 401: the sentence " Sequences specifying the E-gene from SARS-CoV-2, human TNFα induced protein (TSG-6), hepatocyte growth factor 1 (HGF-1), ……… and Staphylococcus aureus. please provide the accession numbers. Also, in materials and methods, these pathogens did not mention (only SARS-CoV-2 and Human breast cancer- or fibroblast), please explain.
Line 447: ".........one of the buffers listed in Table 1A", this should be table 2A instead of 1A.
Line 472: “using ddPCR Supermix for probes (Biorad)…” please, correct Biorad to BioRad.
Did you use a control in your assay? If not, how you can validate your assay?
References: please revise and follow the journal guidelines.
Author Response
Line 14: Abstract: If possible, I suggest providing the most important results in abstract to address which is the best condition of qPCR that defines the “FlashPCR” approach.
We have added the following sentence: “Our approach, called "FlashPCR", uses a protocol involving a 15 second denaturation at 79°C, followed by repeated cycling for 1 second at 79°C and 71°C, together with high Tm primers and specific but simple buffers”.
Line 99: the sentence “P1 and P7 were repeated five times, P4 and P5 four times and P6 was run once”. please provide the reason for that.
We have added this explanation: “An additional repeat for P7 was included to increase our confidence in the repeatability of this, the most extreme protocol. The single run using P6 was due to an accidental mistyping of the polymerisation temperature”.
Line 106: “modified protocols P7 and P8 and buffer 19 to account for instrument-specific variability”, please provide the modification that has been made in M & M.
The modified protocols P7 and P8 were listed in Table 3A, the modified buffer B19 was listed in Table 2A.
Line115: the sentence “the 24- and 26-mer Pentabase primers PF-F and PB-R on denaturation” what do you mean PF-F? is that PB-F?
Yes, typo.
Line 401: the sentence " Sequences specifying the E-gene from SARS-CoV-2, human TNFα induced protein (TSG-6), hepatocyte growth factor 1 (HGF-1), ……… and Staphylococcus aureus.
please provide the accession numbers.
All accession numbers are listed in Table 2B.
Also, in materials and methods, these pathogens did not mention (only SARS-CoV-2 and Human breast cancer- or fibroblast), please explain.
Apologies for omission. Information now included as section 4.7.
Line 447: ".........one of the buffers listed in Table 1A", this should be table 2A instead of 1A.
corrected
Line 472: “using ddPCR Supermix for probes (Biorad)…” please, correct Biorad to BioRad.
corrected
Reviewer 3 Report
Comments and Suggestions for Authors
THE IDEA OF THE PAPER IS TO EVALUATE WAYS TO REDUCE REAL TIME PCR. ESSENTIALLY THESE AUTHORS DEMONSTRATE THAT THEY CAN REDUCE PCR ASSAY TIME DOWN TO ABOUT 15 MIN FROM 30-45 MINUTES. THE PAPER IS WELL WRITTEN AND STRAIGHTFORWARD. I DONT REALLY HAVE ANY CRITISISMS OF THE WORK.
Author Response
Thank you for your positive review. There are no issues to address.
Round 2
Reviewer 1 Report
Comments and Suggestions for Authors
No